

# A lattice pairing-field approach to ultracold Fermi gases

Florian Ehmann[1], Joaquín E. Drut[2] and Jens Braun[1,3]

**1** Institut für Kernphysik, Technische Universität Darmstadt, D-64289 Darmstadt, Germany
**2** Department of Physics and Astronomy, University of North Carolina,
Chapel Hill, North Carolina 27599, USA
**3** ExtreMe Matter Institute EMMI, GSI, Planckstraße 1, D-64291 Darmstadt, Germany

## Abstract

We develop a pairing-field formalism for ab initio studies of non-relativistic two-component fermions on a $(d+1)$-dimensional spacetime lattice. More specifically, we focus on theories where the interaction between the two components can be described by the exchange of a corresponding pairing field. The introduction of a pairing field may indeed be convenient for studies of, e.g., the finite-temperature phase structure and critical behavior of, e.g., ultracold atomic Fermi gases. Moreover, such a formalism allows to directly compute the momentum and frequency dependence of the pair propagator, from which the pair-correlation function can be extracted. For a first illustration of the application of our formalism, we compute the density equation of state and the superfluid order parameter for a gas of unpolarized fermions in $(0+1)$ dimensions by employing the complex Langevin approach to surmount the sign problem.

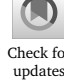

## 1   Introduction

Pair formation of fermions governs the dynamics in a tremendous variety of systems at vastly different length scales, ranging from electrons in solids and ultracold quantum gases to strong-interaction matter and the physics of neutron stars. In gases of interacting spin-1/2 fermions, for example, pairing of spin-up and spin-down particles at diametrically opposite points of the Fermi surface plays a very prominent role, as it leads to the opening of a gap in the quasiparticle spectrum accompanied by the formation of a superfluid state at sufficiently low temperatures [1,2]. Interestingly, such a formation of pairs alongside the emergence of superfluidity even appears to hold for strongly correlated systems, as has been found in detailed theoretical and experimental studies of ultracold Fermi gases (see Refs. [3–24] and Refs. [25–36], respectively). Naturally, an understanding of the microscopic properties of fermion pairs is fundamentally relevant to gain an insight into the macroscopic properties of such fermion gases.

In the present work, for concreteness, we restrict ourselves to systems which can be described by a Hamilton operator of the form

$$\hat{H} = \int \mathrm{d}^d r \left( -\hat{\psi}_\sigma^\dagger(\mathbf{r}) \frac{\nabla^2}{2m_\sigma} \hat{\psi}_\sigma(\mathbf{r}) - g\hat{\psi}_\uparrow^\dagger(\mathbf{r})\hat{\psi}_\uparrow(\mathbf{r})\hat{\psi}_\downarrow^\dagger(\mathbf{r})\hat{\psi}_\downarrow(\mathbf{r}) \right), \tag{1}$$

where we have set $\hbar = 1$ and $d$ determines the number of *spatial* dimensions. Unless stated otherwise, we always assume summation over repeated indices associated with the species $\sigma = \uparrow, \downarrow$. The field operators $\hat{\psi}_\sigma$ and $\hat{\psi}_\sigma^\dagger$ associated with fermions of spin $\sigma$ are assumed to obey the usual anti-commutation relations. We moreover assume that the interaction, parametrized by the coupling $g > 0$, is attractive. From a phenomenological standpoint, we mainly have the development of an alternative lattice formalism for the description of ultracold atomic Fermi gases in mind (which should be extendable to nuclear matter). Still, we would like to add that suitably discretized versions of the model described by Eq. (1) can be directly related to condensed-matter systems, most prominently to the Hubbard model, see, e.g., Refs. [37,38] for recent discussions.

The macroscopic properties of the many-body system described by the Hamilton operator (1) can be extracted from the grand-canonical partition function:

$$Z(\beta, \mu_\uparrow, \mu_\downarrow) = \mathrm{Tr}\exp\left(-\beta\left(\hat{H} - \mu_\uparrow\hat{N}_\uparrow - \mu_\downarrow\hat{N}_\downarrow\right)\right), \tag{2}$$

where $k_\mathrm{B} = 1$ and $\beta = 1/T$ is the inverse temperature. The particle number operators $N_\sigma = \int \mathrm{d}^d r \, \hat{\psi}_\sigma^\dagger(\mathbf{r})\hat{\psi}_\sigma(\mathbf{r})$ couple to the chemical potentials $\mu_\sigma$ associated with the two spin projections.

The partition function can in principle be computed in various ways. In the present work, we restrict ourselves to a path-integral formulation of the partition function. Unfortunately, for an operator of the form (1), exact results are not available for general spatial dimension $d$, temperature $T$, and chemical potentials $\mu_\sigma$, which makes the use of numerical approaches necessary. Instead of computing the path integral directly, however, stochastic path-integral

approaches generally require to remove the Grassmann-valued fermion fields by introducing a suitably chosen auxiliary field. This basically corresponds to performing a Hubbard-Stratonovich transformation [39, 40]. This step is by no means unique. On the contrary, a variety of auxiliary fields have already been used in the literature, see, e.g., Refs. [41–43] for reviews.

Often, the physics of interest suggests the use a specific type of auxiliary field. For example, auxiliary fields closely related to the density are frequently used for computations of the density equation of state and related quantities, see Ref. [43] for a review. However, studies of the propagation of fermion pairs requires the construction of suitable (and in general nontrivial) operators. In such a situation, it may therefore be advantageous to employ an auxiliary field which can be identified with the field associated with such fermion pairs. In particular, for studies of spontaneous symmetry breaking associated with a superfluid state and critical behavior close to the finite-temperature phase boundary, such a pairing-field formulation may be appealing. In the continuum limit, such a formulation indeed exists for studies of, e.g., ultracold Fermi gases, which we shall briefly summarize in Sec. 2, see, e.g., Refs. [44–49] for reviews. This formalism turns out to be particularly convenient for studies of the phase diagram of ultracold Fermi gases, see, e.g., Refs. [6, 17–20, 50]. In Sec. 3, we develop a lattice formulation of this pairing-field formalism. The application of this formulation is then demonstrated in Sec. 4 with the aid of an exactly solvable zero-dimensional fermion model, i.e., the zero-dimensional version of the model described by the Hamilton operator of Eq. (1). We add that our lattice formulation of the well-known continuum formulation of the pairing-field formalism has a sign problem, even for spin-balanced systems, see also Refs. [51,52] for a discussion. This is not unexpected, as the introduction of the pairing field transforms the original purely fermionic model [6, 17–20, 50] into a complex scalar field theory, see our discussion in Secs. 2 and 3. Such theories are known to have a sign problem on a spacetime lattice. In order to deal with this sign problem, we employ the complex Langevin (CL) approach, which has been employed before to study relativistic as well as non-relativistic complex scalar field theories [53–55]. Moreover, the CL approach has been successfully applied to compute properties of ultracold Fermi gases described by the Hamilton operator (1), see Refs. [56–62]. For reviews on the CL approach [63], we refer the reader to, e.g., Refs. [43, 64–68]. In Sec. 4, we discuss this in more detail and also present results for the density equation of state as well as the expectation value of the pairing field. Our conclusions are presented in Sec. 5.

## 2 Pairing field formalism in the continuum limit

For reference, we briefly summarize the derivation of the pairing-field formalism in the continuum limit in this section. It is based on a specific Hubbard-Stratonovich transformation and is well-known in the literature. More detailed discussions can be found in, e.g., Refs. [44–49].

The starting point of our discussion is the path-integral representation of the partition function (2):

$$Z = \int \mathcal{D}(\psi_\sigma^*, \psi_\sigma) \, e^{-S_\mathrm{F}} \,. \tag{3}$$

The (fermionic) action $S_\mathrm{F}$ reads

$$S_\mathrm{F} = \int_0^\beta \mathrm{d}\tau \int \mathrm{d}^d r \left[ \psi_\sigma^* \left( \partial_\tau - \frac{\nabla^2}{2m_\sigma} - \mu_\sigma \right) \psi_\sigma - g \psi_\uparrow^* \psi_\uparrow \psi_\downarrow^* \psi_\downarrow \right], \tag{4}$$

where the fields $\psi_\uparrow$ and $\psi_\downarrow$ are associated with spin-up and spin-down fermions, respectively.

We can now bosonize the theory by performing a Hubbard-Stratonovich transformation [39, 40]. To be more specific, we insert a suitably chosen constant into the path integral.

Here, we choose

$$1 = \int \mathcal{D}(\phi, \phi^*) e^{-g \int_0^\beta \mathrm{d}\tau \int \mathrm{d}^d r\, \phi^*\phi} \,, \tag{5}$$

with a complex-valued auxiliary bosonic field $\phi = \phi(\tau, \mathbf{r})$. We refer to the field $\phi$ as the pairing field and assume that it carries the same quantum numbers as the fermion composite $\psi_\uparrow \psi_\downarrow$. Shifting this field and its complex conjugate according to

$$\phi^* \to \phi^* + \psi_\downarrow^* \psi_\uparrow^*, \quad \text{and} \quad \phi \to \phi + \psi_\uparrow \psi_\downarrow, \tag{6}$$

we arrive at the *paritially bosonized* action:

$$S_{\mathrm{PB}} = \int_0^\beta \mathrm{d}\tau \int \mathrm{d}^d r \left[ \psi_\sigma^* \left( \partial_\tau - \frac{\nabla^2}{2m_\sigma} - \mu_\sigma \right) \psi_\sigma + g\phi^*\phi + g\left( \phi^* \psi_\uparrow \psi_\downarrow - \phi \psi_\uparrow^* \psi_\downarrow^* \right) \right]. \tag{7}$$

From this action we deduce that the complex scalar fields mediate the interaction between the fermions. A resonance in this channel indicates the formation of pairs of spin-up and spin-down fermions which may condense at sufficiently low temperatures. Note also that

$$\langle \phi \rangle = \langle \psi_\downarrow \psi_\uparrow \rangle \,, \tag{8}$$

i.e., the expectation value of the auxiliary field $\phi$ is identical to the expectation value of $\psi_\downarrow \psi_\uparrow$ associated with a pair of fermions, justifying the name *pairing field*. This expectation value is of particular interest as it represents an order parameter for $U(1)$ symmetry breaking as associated with the formation of a superfluid state. Indeed, from Eq. (7), we deduce that a finite expectation value $\langle \phi \rangle$ introduces a gap in the fermion spectrum, as it is the case in standard Bardeen-Cooper-Schrieffer (BCS) theory [1,2].

Since the action (7) is only bilinear in the fermion fields, we can integrate them out to obtain the following path integral:

$$Z = \int \mathcal{D}(\phi, \phi^*)\, e^{-S_{\mathrm{B}}} \,, \tag{9}$$

where we assume that irrelevant normalization factors have been absorbed into the path integral measure. Hence, the bosonic action $S_{\mathrm{B}}$ associated with this path integral reads

$$S_{\mathrm{B}} = -\ln \mathrm{Det}\, M[\phi, \phi^*] + \int_0^\beta \mathrm{d}\tau \int \mathrm{d}^d r\, g\phi^*\phi \,. \tag{10}$$

The fermion matrix $M[\phi, \phi^*]$ appearing in the functional determinant is given by

$$M[\phi, \phi^*] = \begin{pmatrix} \partial_\tau - \frac{\nabla^2}{2m_\uparrow} - \mu_\uparrow & -g\phi \\ -g\phi^* & \partial_\tau + \frac{\nabla^2}{2m_\downarrow} + \mu_\downarrow \end{pmatrix} \delta(\tau - \tau')\delta^{(d)}(\mathbf{r} - \mathbf{r}'). \tag{11}$$

Since the pairing field also appears within this matrix, we are left with a nonlocal bosonic theory. Properties of the pairs of spin-up and spin-down fermions can be studied by computing correlation functions of the pairing field, such the two-point function $\langle \phi(\tau, \mathbf{r})\phi^*(\tau', \mathbf{r}') \rangle$.

We close this section by noting that, at first glance, this fully bosonized theory appears well suited as a starting point for a stochastic evaluation of the path integral. Evaluated on a constant auxiliary field, the action $S_{\mathrm{B}}$ yields nothing but the effective potential in the mean-field approximation, which is indeed real-valued and bounded from below. As we show below, however, the action becomes complex on a spacetime lattice when evaluated on general field configurations, i.e., with a nontrivial dependence on $\tau$ and $\mathbf{r}$.

# 3 Pairing-field formalism on the lattice

## 3.1 Preliminaries

To obtain a lattice formulation of the pairing-field formalism, we begin by replacing the fermionic field operators $\hat{\psi}_\sigma(\mathbf{r})$ in Eq. (1) by corresponding lattice field operators $\hat{\psi}_{\sigma,\mathbf{r}_i}$ which are restricted to a $d$-dimensional (hyper-)cubic lattice of side length $L$ and spacing $a_x$ with periodic boundary conditions. The number of lattice sites in each spatial dimension is given by $N_x = L/a_x$. The Hamilton operator (1) then reads

$$\hat{H} = -\sum_{\mathbf{r}_i} a_x^d \sum_{\mathbf{r}_j} a_x^d \left( \hat{\psi}_{\sigma,\mathbf{r}_i}^\dagger \frac{1}{2m_\sigma} D'_{\Delta,ij} \hat{\psi}_{\sigma,\mathbf{r}_j} \right) - \sum_{\mathbf{r}_i} a_x^d \, g \, \hat{\psi}_{\uparrow,\mathbf{r}_i}^\dagger \hat{\psi}_{\uparrow,\mathbf{r}_i} \hat{\psi}_{\downarrow,\mathbf{r}_i}^\dagger \hat{\psi}_{\downarrow,\mathbf{r}_i}. \tag{12}$$

Here, $\mathbf{r}_i$ determines the position on the spatial lattice and the matrix $D'_\Delta$ represents a finite difference approximation of the continuum Laplace operator. One possible realization of $D'_\Delta$ in one spatial dimension with $\mathbf{r_j} = j a_x \hat{\mathbf{e}}_x$ is given by the central second-order difference:

$$D'_\Delta = \frac{1}{a_x^d a_x^2} \begin{pmatrix} -2 & 1 & & & & 1 \\ 1 & -2 & 1 & & & \\ & & \ddots & & & \\ & & & 1 & -2 & 1 \\ 1 & & & & 1 & -2 \end{pmatrix}. \tag{13}$$

Here and in the following, we only present non-zero entries of matrices. Note that the entries in the top-right corner and the bottom-left corner result from the implementation of periodic boundary conditions. With our concrete choice for the discretization of the Laplace operator specified on the right-hand side of Eq. (13), the sums over spatial lattice sites in Eq. (12) effectively reduce to a sum over all points and their nearest neighbors. However, because said choice for the discretization of the Laplace operator is not unique and may even be optimized such that the convergence towards the continuum limit is improved, we opt to keep the sums in Eq. (12) in their general form.

At this point, we would also like to mention that, for the specific discretization (13) of the Laplace operator, the lattice Hamiltonian is identical to that of a Hubbard model, see, e.g., Refs. [37, 38]. However, exploring this aspect is beyond the scope of the present work.

Following the standard procedure for the derivation of a path-integral expression for the partition function by introducing coherent basis states and a discretization of the imaginary time interval $[0, \beta)$ into $N_\tau$ slices of length $a_\tau = \beta/N_\tau$ (see, e.g., Ref. [69]), we arrive at the following lattice action:

$$\mathcal{S}_{\mathrm{F}} = \sum_{i=1}^{N_\tau} \sum_{\mathbf{r}_j} \left[ -\sum_{\mathbf{r}_k} \psi_{\sigma,\tau_{i+1},\mathbf{r}_j}^* \frac{\bar{D}'_{\Delta,jk}}{2m_\sigma} \psi_{\sigma,\tau_i,\mathbf{r}_k} + \psi_{\sigma,\tau_{i+1},\mathbf{r}_j}^* \left( \psi_{\sigma,\tau_{i+1},\mathbf{r}_j} - (1+\bar{\mu}_\sigma)\psi_{\sigma,\tau_i,\mathbf{r}_j} \right) \right.$$

$$\left. + \bar{g} \, \psi_{\uparrow,\tau_{i+1},\mathbf{r}_j}^* \psi_{\downarrow,\tau_{i+1},\mathbf{r}_j}^* \psi_{\uparrow,\tau_i,\mathbf{r}_j} \psi_{\downarrow,\tau_i,\mathbf{r}_j} \right], \tag{14}$$

with the calligraphic $\mathcal{S}$ distinguishing the lattice action from actions in the continuum. Note that the action $\mathcal{S}_{\mathrm{F}}$ does not depend explicitly on the lattice scales $L$, $a_x$, and $a_\tau$. These quantities as well as the physical parameters $g$, $\beta$, and $\mu_\sigma$ have been absorbed in suitably chosen dimensionless quantities, which also causes the Grassmann-valued field variables $\psi_{\sigma,\tau_i,\mathbf{r}_j}$ (fields evaluated at time $\tau_i$ and position $\mathbf{r}_j$) to be dimensionless. To be specific, we have rescaled the chemical potential with the temporal lattice spacing:

$$\bar{\mu}_\sigma = a_\tau \mu_\sigma = \frac{\beta \mu_\sigma}{N_\tau}. \tag{15}$$

The coupling $g$ has been rescaled and rendered dimensionless as follows:

$$\bar{g} = r^{d/2} N_\tau^{d/2-1} \lambda \,, \tag{16}$$

where $r = a_\tau / a_x^2$ is the dimensionless lattice spacing ratio and $\lambda = \beta^{1-d/2} g$ is the dimensionless coupling. Finally, the dimensionless Laplace operator is given by

$$\bar{D}'_\Delta = r a_x^2 a_x^d D'_\Delta \,. \tag{17}$$

Note that, in our derivation, we are in principle free to choose the specific form of the discretization of the spatial derivatives. However, the discretization of the temporal derivative, specifically the appearance of a backward derivative, follows from the construction of the path integral. In case of relativistic theories, the situation is different as the form of spatial and temporal derivatives is constrained by Lorentz invariance.

We add that the chemical potentials enter our lattice action effectively in the form of constant temporal gauge fields. Indeed, in the continuum limit, a simultaneous transformation of the fermion fields and the chemical potentials exists which leaves the action invariant. This is known as the Silver-Blaze symmetry, see, e.g., Refs. [70–75] for detailed discussions. In our lattice theory, this symmetry is broken by the presence of the temporal lattice and would only be recovered in the continuum limit. In order to preserve this symmetry on the lattice and improve the convergence to the continuum limit, we replace $(1 + \bar{\mu}_\sigma)$ in the discrete temporal derivatives by $e^{\bar{\mu}_\sigma}$, as advocated in Ref. [76]. In practice, this is relevant to obtain accurate results in a regime of small chemical potentials.

## 3.2 Matrix notation

For our development of a lattice pairing-field formalism below, it is convenient to introduce a specific notation for the fields and operators. In this notation, field configurations are represented by column vectors that contain the field values at all lattice sites in an arbitrary but defined order, i.e.,

$$\psi_\sigma = \left( \psi_{\sigma,(\tau,\mathbf{r})_1}, \psi_{\sigma,(\tau,\mathbf{r})_2}, ..., \psi_{\sigma,(\tau,\mathbf{r})_{N_\tau N_x^d}} \right)^{\mathsf{T}} \,, \tag{18}$$

where $\left( (\tau,\mathbf{r})_i \mid i \in \{1,...,N_\tau N_x^d\} \right)$ is used to enumerate all spacetime lattice sites within the box. A scalar product of two such configuration vectors is associated with a spacetime integral over the product of the two fields in the continuum:

$$\psi_\sigma^\dagger \psi_\sigma = \left( \psi_\sigma^* \right)^{\mathsf{T}} \psi_\sigma = \sum_{i=1}^{N_\tau} \sum_{\mathbf{r}_j} \psi_{\sigma,\tau_i,\mathbf{r}_j}^* \psi_{\sigma,\tau_i,\mathbf{r}_j} \,. \tag{19}$$

With the aid of suitably defined matrix multiplications, we can now construct the various terms in the lattice action. For the spatial derivative, we define the matrix $D_\Delta$, which extends the purely spatial matrix $\bar{D}'_\Delta$:

$$\psi_\sigma^\dagger \frac{D_\Delta}{2m_\sigma} \psi_\sigma = \sum_{i=1}^{N_\tau} \sum_{\mathbf{r}_j} \sum_{\mathbf{r}_j} \psi_{\sigma,\tau_i,\mathbf{r}_j}^* \frac{\bar{D}'_{\Delta,jk}}{2m_\sigma} \psi_{\sigma,\tau_i,\mathbf{r}_k} \,. \tag{20}$$

Note that this is not yet the term associated with the spatial derivative which appears in the action (14). In fact, the fermion fields are evaluated at two different points in time in the spatial-derivative term in Eq. (14), which is not the case in Eq. (20). To take into account the

difference in the lattice sites in time direction, we define a (time) *retarder* operator $R_-$ and a (time) *advancer* operator $A_-$, such that

$$\left(A_-\psi_\sigma^*\right)^\mathsf{T}\psi_\sigma = \sum_{i=1}^{N_\tau}\sum_{\mathbf{r}_j}\psi_{\sigma,\tau_i+a_\tau,\mathbf{r}_j}^*\psi_{\sigma,\tau_i,\mathbf{r}_j}\,, \tag{21}$$

and

$$\left(R_-\psi_\sigma^*\right)^\mathsf{T}\psi_\sigma = \sum_{i=1}^{N_\tau}\sum_{\mathbf{r}_j}\psi_{\sigma,\tau_i-a_\tau,\mathbf{r}_j}^*\psi_{\sigma,\tau_i,\mathbf{r}_j}\,. \tag{22}$$

This definition implies that

$$A_-^\mathsf{T} = A_-^{-1} = R_-\,, \tag{23}$$

and that the matrices inherit the temporal boundary conditions of the field. The subscript of these operators refer to boundary conditions: antiperiodic (-) or periodic (+) boundary conditions in time direction. In the present work, we only need those associated with antiperiodic boundary conditions. In the special case of a 0+1-dimensional theory (i.e., $d = 0$), the retarder operator $R_\pm$ has the following matrix representation:[1]

$$R_\pm = \begin{pmatrix} 0 & & & \pm 1 \\ 1 & 0 & & \\ & \ddots & \ddots & \\ & & 1 & 0 \end{pmatrix}\,. \tag{24}$$

The retarder and advancer operators allow us to define discrete temporal derivatives as

$$D_\tau^{(\mathrm{bw})}(\bar{\mu}_\sigma) = \mathbb{1} - e^{\bar{\mu}_\sigma}R_-\,, \tag{25}$$

and

$$D_\tau^{(\mathrm{fw})}(\bar{\mu}_\sigma) = e^{\bar{\mu}_\sigma}A_- - \mathbb{1}\,, \tag{26}$$

It follows that

$$D_\tau^{(\mathrm{fw})}(\bar{\mu}_\sigma) = -\left(D_\tau^{(\mathrm{bw})}(\bar{\mu}_\sigma)\right)^\mathsf{T}\,, \tag{27}$$

where we have used Eq. (23).

For the interaction term, we introduce the vectors $(\psi_\uparrow \circ \psi_\downarrow)$ and $(\psi_\uparrow^* \circ \psi_\downarrow^*)$ with $\circ$ being the element-wise Hadamard product. With this notation at hand, we can rewrite our discrete action (14) as follows

$$\mathcal{S}_\mathrm{F} = \psi_\sigma^\dagger D_\tau^{(\mathrm{bw})}(\bar{\mu}_\sigma)\psi_\sigma + \psi_\sigma^\dagger \frac{R_- D_\Delta}{2m_\sigma}\psi_\sigma - \bar{g}(\psi_\uparrow^* \circ \psi_\downarrow^*)^\mathsf{T}R_-(\psi_\uparrow \circ \psi_\downarrow)\,. \tag{28}$$

This form of the fermionic action represents the starting point for our development of a lattice pairing-field formalism in the next subsection.

## 3.3 The discretized pairing field

Analogously to the continuum theory, we now perform a Hubbard-Stratonovich transformation to eventually remove the fermionic degrees of freedom. To this end, we again insert a constant factor into the path integral expression of the partition function. We choose

$$1 = \int \mathcal{D}(\phi^*,\phi)e^{-g\phi^\dagger\phi}\,, \tag{29}$$

---

[1]Here, we tacitly assume that the field values in the field-configuration vector are ordered according to their time coordinate.

where

$$\phi^\dagger \phi = \sum_{i=1}^{N_\tau} \sum_{\mathbf{r}_j} \phi^*_{\tau_i,\mathbf{r}_j} \phi_{\tau_i,\mathbf{r}_j}. \tag{30}$$

However, in contrast to the continuum, we cannot perform a shift like

$$\phi^* \to \phi^* - (\psi^*_\uparrow \circ \psi^*_\downarrow), \quad \text{and} \quad \phi \to \phi + (\psi_\uparrow \circ \psi_\downarrow), \tag{31}$$

since due to the time shifting matrix $R_-$ in the interaction term in the action in Eq. (28) this shift will not create a term that cancels the four-fermion interaction. To mitigate this, one might be tempted to begin the Hubbard-Stratonovich transformation by inserting a different factor of one already containing the time shift, i.e.,

$$1 = \int \mathcal{D}(\phi^*, \phi) e^{-g\phi^\dagger R_- \phi}, \tag{32}$$

however, since the time shifting matrices are not positive-definite, such a Hubbard-Stratonovich transformation would not be well defined.

Instead, we shift the starred and unstarred bosonic fields independently, i.e.,

$$\phi^* \to \phi^* - A_-(\psi^*_\uparrow \circ \psi^*_\downarrow), \quad \text{and} \quad \phi \to \phi + (\psi_\uparrow \circ \psi_\downarrow), \tag{33}$$

and then obtain the partially bosonized lattice action:

$$\mathcal{S}_{\text{PB}} = \psi^\dagger_\sigma D^{(\text{bw})}_\tau(\bar\mu_\sigma)\psi_\sigma - \psi^\dagger_\sigma \frac{R_- D_\Delta}{2m_\sigma}\psi_\sigma + g\phi^\dagger\phi + g\left(\phi^\dagger(\psi_\uparrow \circ \psi_\downarrow) - (\psi^*_\uparrow \circ \psi^*_\downarrow)^\mathsf{T} R_- \phi\right). \tag{34}$$

As in the continuum limit, the fermions can now be integrated out to obtain the bosonized lattice action:[2]

$$\mathcal{S}_{\text{B}} = -\ln\det\mathcal{M} + \bar{g}\phi^\dagger\phi, \tag{35}$$

where the fermion matrix $\mathcal{M}$ is given by

$$\mathcal{M} = \begin{pmatrix} D^{(\text{bw})}_\tau(\bar\mu_\uparrow) - \frac{R_- D_\Delta}{2m_\uparrow} & -\bar{g}\,\text{Diag}(R_-\phi) \\ -\bar{g}\,\text{Diag}(\phi^*) & D^{(\text{fw})}_\tau(\bar\mu_\downarrow) + \frac{A_- D_\Delta}{2m_\downarrow} \end{pmatrix}. \tag{36}$$

Note that this matrix is in general not positive-definite and therefore the bosonized action is in general complex, which leads to a sign problem in conventional stochastic computations of the path integral, even in the limit of vanishing spin and mass imbalances.[3]

---

[2] This can be done by reordering terms of the form $\psi^\dagger_\downarrow \mathcal{O}\psi_\downarrow$ such that they appear as terms of the form $\psi^\mathsf{T}_\downarrow \mathcal{O}'\psi^*_\downarrow$ in the partially bosonized action which allows to rewrite the path integral as a Gaussian integral in the fermionic degrees of freedom. In the continuum limit, the determination of the new operator $\mathcal{O}'$ requires an integration by parts. In our lattice formalism, this is done by transposition. For example, we have

$$\psi^\dagger_\downarrow D^{(\text{bw})}_\tau(\bar\mu_\downarrow)\psi_\downarrow = \left(\psi^\dagger_\downarrow D^{(\text{bw})}_\tau(\bar\mu_\downarrow)\psi_\downarrow\right)^\mathsf{T}$$
$$= -\psi^\mathsf{T}_\downarrow D^{(\text{fw})}_\tau(\bar\mu_\downarrow)\psi^*_\downarrow.$$

The fermions can then be conveniently integrated out by eventually introducing Nambu-Gorkov spinors $\psi^\dagger = (\psi^\dagger_\uparrow, \psi^\mathsf{T}_\downarrow)$ and $\psi = (\psi_\uparrow, \psi^*_\downarrow)^\mathsf{T}$.

[3] There also exist Hubbard-Stratonovich transformations which do not suffer from a sign problem, as long as masses and chemical potentials of the two fermion species are identical. An example of such a formulation would be the density formulation discussed in, e.g., Ref. [42]. This approach comes with a computationally less costly update process in numerical applications, at the cost of a more elaborate calculation of pairing observables, such as the superfluid order parameter and the spacetime-dependent pair propagator, see also Sec. 2. The latter quantities are directly accessible with our pairing-type Hubbard-Stratonovich transformation. As a side effect, it also creates a closer connection to continuum calculations where the use of a pairing field as auxiliary field is more common.

Let us finally add that our partially bosonized lattice action $\mathcal{S}_{\mathrm{PB}}$ and its continuum analogue in Eq. (7) look similar at first glance. However, we would like to emphasize that a naive discretization of the continuum theory does not lead us to our lattice theory. For example, a naive discretization of the continuum theory in Eq. (7) would lead to a Yukawa interaction term which differs from the partially bosonized lattice action $\mathcal{S}_{\mathrm{PB}}$ by

$$(\psi_\uparrow^* \circ \psi_\downarrow^*)^\intercal \phi - (\psi_\uparrow^* \circ \psi_\downarrow^*)^\intercal R_- \phi \sim \mathcal{O}(a_\tau). \tag{37}$$

This indicates that the difference in the interaction term vanishes in the limit $a_\tau \to 0$, which also holds for the kinetic terms. In fact, in the continuum limit $N_\tau \to \infty$ and $N_x \to \infty$ (such that $N_\tau a_\tau = \beta$ and $N_x a_x = L$ remain constant), the finite difference operators become derivatives and, loosely speaking, the effect of the retarder and advancer matrices vanishes. The discrete theory then indeed becomes the continuum theory given in Eq. (7).

Finally, we would like to state again that the specific type of Hubbard-Stratonovich transformation underlying our formalism is not new but has been frequently employed in continuum studies, see, e.g., Refs. [44–49]. Here, we have derived a lattice formulation of this approach. Let us also add that a large variety of Hubbard-Stratonovich transformations has been discussed in the literature to study models as described by the Hamilton operator (1). In this respect, we emphasize that our present lattice formalism should not be confused with the one considered in Ref. [77], where a density-type field is used as an auxiliary field and homogeneous pairing fields are only introduced via source terms. Moreover, it should be mentioned that, in contrast to the spacetime representation of the auxiliary field used in our present work, lattice formulations built on a purely spatial representation of the problem are also very popular. The latter also include a formulation, where the auxiliary field is introduced by coupling it to the pairing operators [51]. In this context, it is worth mentioning that the sign problem has also been studied on more general grounds based on a symmetry classification of theories [78–81]. However, an analysis of whether and how the classifications presented in these works can be applied to our spacetime formulation of theories with an even number of fermion species is beyond the scope of our present study. In any case, our analysis of the fermion matrix (36) indicates that our pairing-field approach comes with a sign problem, in accordance with, e.g., Ref. [52]. For recent reviews on the sign problem and the role of Hubbard-Stratonovich transformations in the context of stochastic calculations, we refer the reader to Refs. [42, 68, 82].

## 3.4 Lattice parameters

Based on the required range and accuracy of the results, we can define criteria that determine the lattice parameters. In the following, we present criteria for the determination of the numbers of lattice sites $N_\tau$ and $N_x$ based on the largest chemical potential $\beta \mu_{\mathrm{max}}$ of interest and three empirical constants $\delta_{\mathrm{SB}}$, $C_I$ and $C_\lambda$. The latter two control the numerical accuracy of the results. For a given $\beta \mu_{\mathrm{max}}$, the numerically exact solution is approached in the limit $C_I \to \infty$ and $C_\lambda \to \infty$, wherein the vanishing deviation caused by the replacement term to restore the Silver-Blaze symmetry is ensured by the increasing temporal lattice size following the limit for $C_I$.

### 3.4.1 Number of temporal lattice sites

By considering the energy scale set by the interaction and the Silver-Blaze symmetry, we first explore criteria which have to be satisfied by the temporal lattice (in terms of extent and spacing). For our numerical studies presented in Sec. 4, we have in general used the smallest even value of $N_\tau$ which fulfills all the criteria. However, we have also checked the correctness

of our criteria by studying the convergence of the density as a function of $N_\tau$ for selected parameter sets.

On a spacetime lattice, the reciprocal temporal spacing defines a cutoff for the energies that can be resolved on the lattice. We therefore must ensure that the energy scale $E_I$ set by the interaction is sufficiently small compared to this cutoff, i.e., $1/a_\tau > C_I E_I$ with an empirically determined factor $C_I$. Taking into account the dimension of the coupling parameter $g$, we define the interaction energy scale $E_I = \lambda a_x^{-d} \beta^{d/2-1}$. For $d < 2$ this leads us to the criterion

$$N_\tau > \left( \lambda r^{d/2} C_I \right)^{\frac{1}{1-d/2}} . \tag{38}$$

In the special case of $d = 2$, the coupling is dimensionless and does not provide an energy scale. For $d > 2$, the scale set by the interaction energy imposes an upper bound for the number of temporal lattice sites.

In the derivation of our formalism, we replaced factors of $(1 + \bar{\mu})$ by factors of $e^{\bar{\mu}}$ to preserve the Silver-Blaze symmetry. Such an improved lattice theory leads to faster convergence towards the continuum theory relative to the unimproved counterpart. Specifically, the difference between improved and unimproved increases quadratically at leading order with increasing values of $|\bar{\mu}|$. Since we have

$$\bar{\mu} = \frac{\beta \mu}{N_\tau} , \tag{39}$$

we can reduce the deviation by simply increasing the temporal lattice extent $N_\tau$. In practice, we define a range absolutes of values of interest for the quantity $\beta\mu$. The upper bound of this range, $\beta\mu_{\max}$, is then used to determine an appropriate value of $N_\tau$ by solving the equation:

$$e^{\beta\mu_{\max}/N_\tau} - \left( 1 + \frac{\beta\mu_{\max}}{N_\tau} \right) < \delta_{\mathrm{SB}} . \tag{40}$$

This relation sets an empirical upper bound $\delta_{\mathrm{SB}}$ for the difference. The results in this work are obtained with $\delta_{\mathrm{SB}} = 0.005$. However, for the coupling strengths considered in Sec. 4, the temporal lattice spacing is determined by the scale set by the interaction energy rather than the Silver-Blaze correction.

### 3.4.2 Number of spatial lattice sites

In addition to the temporal lattice size, we have to determine a number of spatial lattice sites. In this case, we have to ensure that the length of our box $L = a_x N_x$ is sufficiently large compared to the thermal wavelength $\lambda_{\mathrm{th},\sigma} = (2\pi\beta/m_\sigma)^{1/2}$. To this end, we introduce an empirical constant $C_\lambda$ and require

$$C_\lambda \lambda_{\mathrm{th},\sigma} \leq a_x N_x , \tag{41}$$

Note that the (inverse) temperature is a phenomenological control parameter rather than an artificial parameter of our lattice theory, but, contrary to that, the spatial extent $L$ of our (hypercubic) lattice is in general an artificial parameter which has been introduced to evaluate the path integral numerically. That being stated, the above inequality leads us to the criterion

$$N_x \geq C_\lambda \left( \frac{2\pi r}{m_\sigma} N_\tau \right)^{1/2} , \tag{42}$$

which determines the value of $N_x$ based on the number of temporal lattice sites $N_\tau$. In other words, this inequality relates the temporal lattice spacing to the spatial lattice spacing for given values of $\beta$ and $L$. We also note, that the required number of spatial lattice sites per dimension scales with the square root of the number of temporal lattice sites, as $N_x \propto N_\tau^{1/2}$.

# 4 Application: Fermions in zero dimensions

To illustrate and discuss the application of our lattice pairing-field formalism, we consider our model in the limit of zero spatial dimensions ($d = 0$). This is motivated by the fact that exact analytic results in closed form can be obtained for at least some physical observables in this case, providing a clean benchmark for our numerical calculations.

## 4.1 Exact analytic results

The partition function (2) of our model can be computed analytically for $d = 0$. We find

$$Z = (1 + e^{\beta \mu_\uparrow})(1 + e^{\beta \mu_\downarrow}) + e^{\beta(\mu_\uparrow + \mu_\downarrow)}(e^{\beta g} - 1). \tag{43}$$

From this formula, we can extract exact results for the spin-up and spin-down densities by taking a derivative with respect to $\mu_\uparrow$ and $\mu_\downarrow$, such that

$$n_\sigma = \frac{e^{\beta \mu_\sigma} + e^{\beta(\mu_\uparrow + \mu_\downarrow + g)}}{(1 + e^{\beta \mu_\uparrow})(1 + e^{\beta \mu_\downarrow}) + e^{\beta(\mu_\uparrow + \mu_\downarrow)}(e^{\beta g} - 1)}. \tag{44}$$

The total density is defined to be the sum of the spin-up and spin-down density: $n = n_\uparrow + n_\downarrow$. Below, we often use the densities of the noninteracting system to normalize the densities of the interacting system. We have

$$n_{0,\sigma} := n_\sigma \big|_{g=0} = \frac{1}{1 + e^{-\beta \mu_\sigma}}, \tag{45}$$

and $n_0 := n_{0,\uparrow} + n_{0,\downarrow}$. We add that the densities of the two fermion species become independent of the chemical potentials in the limit of an infinitely strong attractive interaction:

$$n_\sigma \big|_{\beta g \to \infty} \to 1. \tag{46}$$

Moreover, for $\mu_\uparrow = \mu_\downarrow = 0$ and $g > 0$, the normalized densities $n_\sigma / n_0$ are found to be bounded from below and above: $1 \leq n_\sigma / n_{0,\sigma} \leq 2$. Therefore, this ratio may be considered a macroscopic measure of the effective strength of the interactions in the system. Note that, for infinite repulsion, the densities still depend on the chemical potentials.

The main focus of our illustrative numerical studies will be on the computation of densities. Of course, the actual strength of our formalism lies in the computation of observables which can be constructed directly from the pairing field, such as the pair-correlation function, the quasiparticle gap as well as other order parameters for superfluidity, which can in principle also be computed analytically in $d = 0$. We will report on such computations elsewhere and only show results for the expectation value of the pairing field in this work. From an analytic calculation of the latter quantity, we obtain $\langle \phi \rangle = 0$ since

$$\langle \psi_\downarrow \psi_\uparrow \rangle = 0, \tag{47}$$

and $\langle \phi \rangle = \langle \psi_\downarrow \psi_\uparrow \rangle$. Note that $\langle \phi \rangle$ can be used as an order parameter for $U(1)$ symmetry breaking in our model. This allows to relate this quantity to the formation of a superfluid ground state in $d \geq 2$ in the long-range limit.

We close this subsection with a comment on mean-field theory. It is also possible to study our zero-dimensional model in the mean-field approximation in the continuum limit. The effective potential $U$ then reads

$$U = g \bar{\phi}^* \bar{\phi} - \frac{1}{\beta} \ln \left( \cosh(\beta h) + \cosh \left( \beta \sqrt{\mu^2 + |\phi|^2} \right) \right), \tag{48}$$

where $\mu$ is the average chemical potential of the two species,

$$\mu \equiv \frac{\mu_\uparrow + \mu_\downarrow}{2}, \tag{49}$$

and the chemical potential asymmetry is measured in terms of the so-called Zeeman field $h$,

$$h \equiv \frac{\mu_\uparrow - \mu_\downarrow}{2}. \tag{50}$$

The fields $\bar{\phi}$ and $\bar{\phi}^*$ represent constant pairing fields. Note that, in $U$, we dropped terms independent of the field $\phi$ and its complex conjugate.

A minimization of the effective potential $U$ yields the ground state $\bar{\phi}_0$. The pressure equation of state can be extracted from an evaluation of $U$ at the ground state $\bar{\phi}_0 = \langle \psi_\downarrow \psi_\uparrow \rangle$. From the first derivative of the pressure with respect to the chemical potential $\mu_\sigma$, we then obtain the densities. Notably, for the coupling strengths considered in the present work, the mean-field approximation yields that the density in the interacting system is identical to the one in the noninteracting system, even if our numerical studies (in accordance with the exact results) predict that this is not the case. Even more, for sufficiently attractive couplings (beyond those considered in our numerical studies below), the mean-field approximation indicates that the ground state is nontrivial, i.e., $\bar{\phi}_0 = \langle \psi_\downarrow \psi_\uparrow \rangle \neq 0$, again in disagreement with the exact solution. Thus, the results from the mean-field approximation are not even correct on a qualitative level. However, this is not unexpected as fluctuation effects (which are missing in the mean-field approximation) are very relevant in low-dimensional systems.

## 4.2 Numerical framework

### 4.2.1 Complex Langevin approach

Since we have integrated out the fermion fields in our formalism, we are left with a purely bosonic field theory. As such, the dynamics of the system is fully encoded in the pairing field $\phi$ and its complex conjugate $\phi^*$. In order to compute observables with the path integral, numerically it is convenient to generate a finite number suitable pairing-field configurations which allow us to evaluate the path integral in an efficient way by approximating its value with the average of the integrand at the chosen field configurations. Often this is done by using Monte-Carlo (MC) techniques. Unfortunately, standard MC techniques are not applicable in our case since the bosonized lattice action $\mathcal{S}_B$ is complex, as already indicated above. Therefore, we employ the CL approach [63] to generate field configurations suitable for an efficient computation of observables, see Refs. [43, 64–68] for reviews.

The application of the CL approach requires a complexification of the fields in our bosonized theory. To this end, we first rewrite the pairing field $\phi$ and its complex conjugate $\phi^*$ in terms of their real and imaginary parts which leaves us with two real-valued fields, $\phi_1$ and $\phi_2$:

$$\phi = \phi_1 + i\phi_2, \tag{51}$$

where $\phi_1 := \mathrm{Re}(\phi)$ and $\phi_2 := \mathrm{Im}(\phi)$. On the lattice, these fields are parametrized by a finite set of complex numbers as described above in our derivation of the lattice action $\mathcal{S}_B$:

$$\phi_k(\tau, \mathbf{r}) \to \left\{ \phi^{(k)}_{\tau_i, \mathbf{r}_j} \right\}, \tag{52}$$

where $k = 1, 2$. Within our CL approach, these real-valued fields are then complexified, i.e., the field $\phi_1$ and $\phi_2$ are considered to be complex fields. Formally, this is achieved by replacing these fields by a sum of their real and imaginary parts. For the field variables, this replacement reads

$$\phi^{(k)}_{\tau_i, \mathbf{r}_j} \to \tilde{\phi}^{(k)}_{\tau_i, \mathbf{r}_j} = \tilde{\phi}^{(k,1)}_{\tau_i, \mathbf{r}_j} + i \tilde{\phi}^{(k,2)}_{\tau_i, \mathbf{r}_j}. \tag{53}$$

Note that this complexification of the fields implies that we have to deal with four real-valued fields when we compute physical quantities.

In addition to the complexification of the original field variables, the CL approach also introduces a fictitious time $t_{\text{CL}}$. The complexified fields are then evolved along this so-called CL time according to the following set of discretized differential equations for our complexified fields:

$$\tilde{\phi}_{\tau_i, \mathbf{r}_j}^{(k)}(n+1) = \tilde{\phi}_{\tau_i, \mathbf{r}_j}^{(k)}(n) + \delta t_{\text{CL}} K_{\tau_i, \mathbf{r}_j}^{(k)}(n) + \sqrt{\delta t_{\text{CL}}}\, \eta_{\tau_i, \mathbf{r}_j}^{(k)}(n), \tag{54}$$

where $n$ is the CL time index of the fields and $\delta t_{\text{CL}}$ is the CL time step, $t_{\text{CL}} = n\,\delta t_{\text{CL}}$. The noise $\eta$ is real and Gaussian with

$$\langle \eta_{\tau_i, \mathbf{r}_j}^{(k)}(n) \rangle = 0, \tag{55}$$

and

$$\langle \eta_{\tau_{i_1}, \mathbf{r}_{j_1}}^{(k_1)}(n_1) \eta_{\tau_{i_2}, \mathbf{r}_{j_2}}^{(k_2)}(n_2) \rangle = 2\delta_{n_1, n_2} \delta_{k_1, k_2} \delta_{i_1, i_2} \delta_{j_1, j_2}. \tag{56}$$

The so-called drift term is defined as

$$K_{\tau_i, \mathbf{r}_j}^{(k)}(n) = -\left. \frac{\partial \mathcal{S}_{\text{B}}}{\partial \phi_{\tau_i, \mathbf{r}_j}^{(k)}} \right|_{\{\phi \to \tilde{\phi}(n)\}}. \tag{57}$$

This set of coupled differential equations can now be used to generate field configurations for the evaluation of physical observables in the spirit of the original path-integral formulation. For more details on the CL approach in the context of nonrelativistic many-body physics, we refer the reader to Ref. [43].

### 4.2.2 Computation of observables

The computation of a physical observable requires the definition of a corresponding operator in our formalism. For example, we can derive an operator $\mathcal{O}_{N_\sigma}^{\text{CL}}$ for the calculation of the particle number $N_\sigma$ of species $\sigma$ from the definition of the partition function:

$$\begin{aligned} N_\sigma &= \frac{1}{Z} \text{Tr}\, \hat{N}_\sigma e^{-\beta(\hat{H} - \mu_\uparrow \hat{N}_\uparrow - \mu_\downarrow \hat{N}_\downarrow)} \\ &= \frac{1}{Z} \frac{1}{\beta} \partial_{\mu_\sigma} \int \mathcal{D}(\phi^*, \phi)\, e^{-\mathcal{S}_{\text{B}}} \\ &= \frac{1}{N_\tau} \left\langle \text{tr}\left( (\partial_{\tilde{\mu}_\sigma} \mathcal{M}) \mathcal{M}^{-1} \right) \right\rangle, \end{aligned} \tag{58}$$

where $\mathcal{M}$ is the fermion matrix introduced in Eq. (36). From this, we deduce the following "CL representation" of the density operator:

$$\mathcal{O}_{N_\sigma}^{\text{CL}}(\{\tilde{\phi}_{\tau_i, \mathbf{r}_j}^{(k)}(n)\}) = \frac{1}{N_\tau} \text{tr}\left( (\partial_{\tilde{\mu}_\sigma} \mathcal{M}) \mathcal{M}^{-1} \right) \Big|_{\{\phi \to \tilde{\phi}(n)\}}. \tag{59}$$

Computing the average of this operator with the field configuration samples obtained from the solution of the CL equations, we obtain the particle number of the species $\sigma$. To be specific, we have

$$N_\sigma = \frac{1}{N_{\text{CL}}} \sum_{n=1}^{N_{\text{CL}}} \mathcal{O}_{N_\sigma}^{\text{CL}}(\{\tilde{\phi}_{\tau_i, \mathbf{r}_j}^{(k)}(n)\}), \tag{60}$$

where $N_{\text{CL}}$ is the number of CL time steps.

Since our conventions for all quantities appearing in the action $\mathcal{S}_{\text{B}}$ are such that our formalism is completely free of any dimensionful scale, we can strictly speaking only extract dimensionless observables from our CL calculations. Since the density $n = N/V$ depends on the

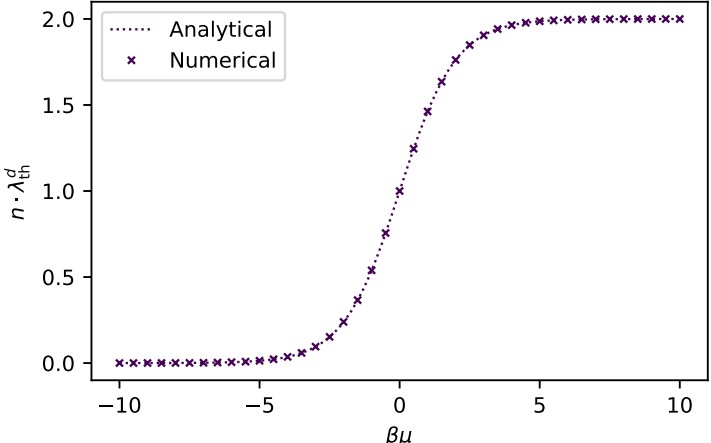

Figure 1: Dimensionless density for a non-interacting system, i.e. $\lambda = 0$. The numerical results are in perfect alignment with the analytical solution.

volume of the box, $V = (N_x a_x)^d$, it can therefore not be computed directly. Instead, we define a dimensionless density by multiplying the density $n$ with the thermal wavelength volume $\lambda_{\text{th}}^d$:

$$n\lambda_{\text{th}}^d = N\frac{\lambda_{\text{th}}^d}{V} = N\left(\frac{2\pi r N_\tau}{N_x^2}\right)^{d/2}. \tag{61}$$

Here, the thermal wavelength $\lambda_{\text{th}} = (2\pi\beta)^{1/2}$ is made species-independent by using a mass of one. Note that in 0+1 dimensions the physical density $n$ and the dimensionless density $n\lambda_{\text{th}}^d$ are identical as the theory has no spatial extent.

Finally, we would like to add that the computation of expectation values of the pairing fields, such as the superfluid order parameter and the pair propagator, can be implemented straightforwardly since the pairing field is the fundamental field in our formalism. For example, we have

$$\begin{aligned}\langle\phi_{\tau_i,\mathbf{r}_j}\rangle &= \frac{1}{Z}\int\mathcal{D}(\phi^*,\phi)\,\phi(\tau_i,\mathbf{r}_j)\,e^{-S_{\text{B}}} \\ &\approx \frac{1}{N_{\text{CL}}}\sum_{n=1}^{N_{\text{CL}}}\left(\tilde\phi_{\tau_i,\mathbf{r}_j}^{(1)}(n) + i\tilde\phi_{\tau_i,\mathbf{r}_j}^{(2)}(n)\right).\end{aligned} \tag{62}$$

Expectation values of more than one pairing field can be computed correspondingly. Note that the computation of fermionic correlation functions (e.g., the propagator of one of the fermion species) is more involved as it requires to introduce a source into the path integral.

## 4.3 Results for the density equation of state

For $\lambda = 0$, we simply expect to obtain the density distribution of a non-interacting Fermi gas for each of the species. Indeed, as Fig. 1 shows, the simulation reproduces our expectation precisely. No error bars are shown in this figure because, without interaction, the density is independent of the pairing field. Therefore, no sampling is needed, and the calculation of the observable is numerically exact.

For the interacting case we study the density in units of the non-interacting density $n_0$, i.e., $n/n_0$ in the balanced case of $\mu_\uparrow = \mu_\downarrow \equiv \mu$. In Fig. 2, we show densities for a range of interaction strengths, together with the corresponding analytical results. The error bars

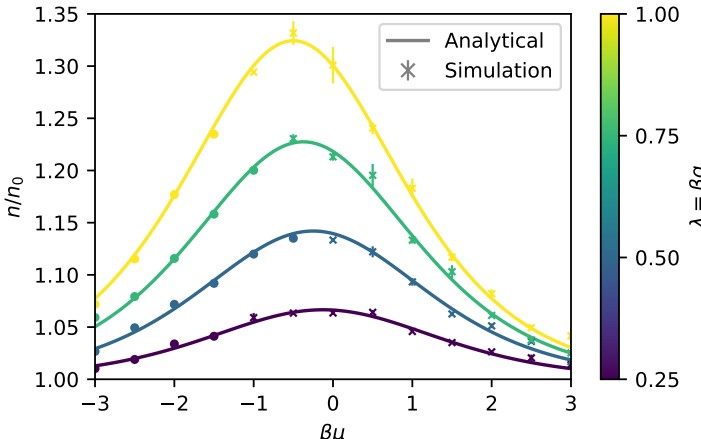

Figure 2: Ratio of interacting and non-interacting density for interacting systems with $\lambda \in \{0.25, 0.5, 0.75, 1.0\}$. The errorbars represent the statistical uncertainty of the data points. For the data points marked with an "x" lattice effects are extrapolated out, while those marked with an "o" are obtained using a maximum likelihood estimator.

represent an estimate of the standard deviation of the MC samples obtained through Jackknife resampling [83] to reduce effects of autocorrelation in the samples. For chemical potentials $\beta\mu \gtrsim -1$, the calculated densities start to exhibit a significant dependence on the parameter $C_I$ which determines the size of the lattice. To remove these lattice artefacts, we have performed an extrapolation of the data in this regime. These lattice artefacts are extrapolated out in this $\beta\mu$-regime. For $\beta\mu \lesssim -1$, the dependence of the calculated densities on the lattice size is negligible and a maximum likelihood estimator was used to determine the density instead. We refer the reader to App. A for a more detailed discussion.

We observe that that the calculated densities are in excellent agreement with the analytic solution. We emphasize that this is non-trivial. For example, in the mean-field approximation, the density equation of state agrees identically with the one of the non-interacting gas in this coupling regime, i.e., $n/n_0 = 1$ for all values of $\beta\mu$.[4] Interestingly, we find that the observed dependence of the density on $\beta\mu$ is qualitatively very similar to the one found in one- and two-dimensional fermion models of this type, see, e.g., Refs. [58, 84–86]. Note that, in the unitary limit (in three dimensions), the density equation of state increases monotonically as a function of $\beta\mu$ and does not develop a maximum around $\beta\mu = 0$, see, e.g., Refs. [87, 88].

## 4.4 Pairing

In principle, we can also search for non-trivial ground state configurations in terms of the pairing field. For example, in Subsec. 4.1, we have computed the effective potential $U$ in the mean-field approximation which develops a non-trivial ground state associated with $\langle \phi \rangle \neq 0$ for $\beta g > 4$. Such a non-trivial ground state would be associated with spontaneous symmetry breaking and the formation of superfluid gap. Of course, in the present zero-dimensional model, the appearance of a superfluid ground state is simply a failure of the mean-field approximation. In the following we are not interested in a demonstration of the breakdown of the mean-field approximation in this strong-coupling regime. We rather aim at a demonstration that expectation values of the pairing field (or even products of multiple pairing fields as

---

[4]This eventually follows from a minimization of the effective potential $U$ in the mean-field approximation, see Eq. (48).

associated with correlation functions) are indeed directly accessible in our approach. To be specific, we can study the behavior of the pairing field throughout the MC sampling process by evaluating the observable $\tilde{\phi}$,

$$\tilde{\phi} = \lambda_{\text{th}}^{d/2} \langle \phi \rangle_{\tau, \mathbf{r}} = \frac{\lambda_{\text{th}}^{d/2}}{\beta N_x^d a_x^d} \int_0^\beta d\tau \int_{\mathbf{r}} d^d r \, \phi(\tau, \mathbf{r}), \tag{63}$$

which can be calculated from the dimensionless discrete field configuration vector $\bar{\phi}$ as

$$\tilde{\phi}_{\text{sample}} = \frac{(2\pi r N_\tau)^{d/4}}{N_\tau N_x^d} \sum_i \bar{\phi}_i. \tag{64}$$

Since we have $\langle \phi \rangle = \langle \psi_\downarrow \psi_\uparrow \rangle$, i.e., the expectation value of two annihilation operators, we expect to find a sample mean of zero for the observable $\tilde{\phi}$.

In Fig. 3, we show the sampling process for the observable $\tilde{\phi}$ for one simulation run. As it should be, we find that the mean of $\tilde{\phi} = 0$ lies within the deviation of the result, already without correcting systematic sources of uncertainty (which originate from the CL time spacing and the lattice size).

Beyond the expected expectation value of $\tilde{\phi}$, we can also see that the sampling process is well behaved and does neither "get stuck" nor exhibit "large jumps" that can occur when singularities are present in the drift term of the CL equation. Such singularities in the drift indeed form a potentially serious impediment for the CL method. A review of this aspect can be found in Ref. [43], including an illustration of some of the problems emerging from the presence of singularities in the drift term with the aid of a $0+0$-dimensional theory (i.e., a theory where the fields depend neither on time nor space). In our present work, however, we consider a spacetime formulation of the problem, such that even the $0+1$-dimensional case considered in our numerical studies already features high-dimensional integrals, rendering an exact analysis of singularities in the drift (as done for the 0+0-dimensional case in Ref. [43] and also in various ways in Refs. [89–94]) impractical. We therefore only analyze the sampling process, which we find to be localized to a region around the origin in the field space, and use the indicator observable defined in Eq. (63) to spot potential singularities. We find that the drift indeed seems to be free of singularities in our present study, as already indicated above. However, for studies with a finite number of space dimensions, the issues associated with singularities in the drift may have to be carefully analyzed on a case-by-case basis. Note that the localization of the sampling process in our formulation also aides us performing the sampling process more efficiently, indicating that the theory is a good candidate for treatment with the CL method.

## 4.5 Sign problem

Our system of interest exhibits a sign problem, even in the absence of a spin or mass imbalance, which follows directly from the properties of the matrix $\mathcal{M}$ in Eq. (36) when evaluated on non-uniform field configurations. To surmount this sign problem, we have chose the CL approach to MC sampling. As a consequence of that, however, both the real and imaginary part of the pairing field are themselves complex quantities. This results in a complex integrand of the path integral for all interacting systems. In this subsection, we would like to briefly study this aspect by analyzing the MC samples of the weight of the path integral $e^{-S_{\text{B}}}$.

In Fig. 4, we present distribution histograms of the complex argument of the path integral weight $e^{-S_{\text{B}}}$ for different values for the chemical potential $\beta\mu$ and the interaction $\lambda$. We generally observe narrow distributions around an argument of zero. Thus, the weights are close to the real axis but exhibit a weak phase problem. The width of the phase distribution increases

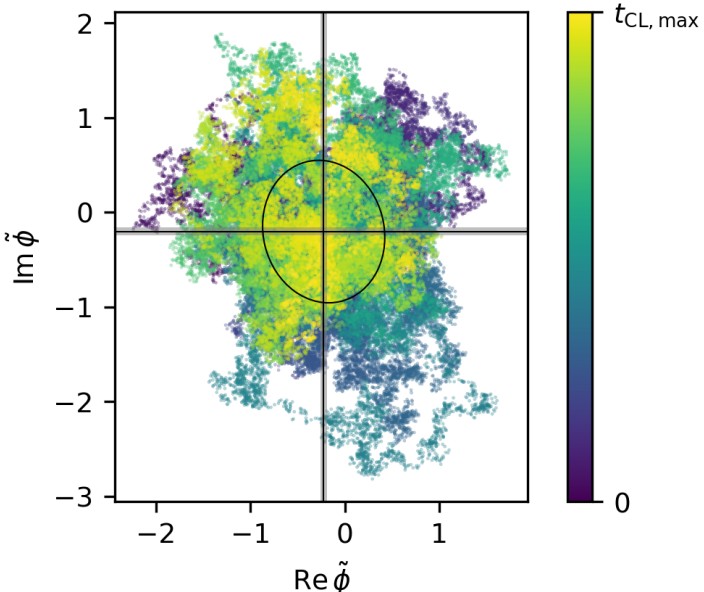

Figure 3: MC samples points of $\tilde{\phi}$ in the complex plane. The color indicates at which CL time the sample point is calculated; the simulation ends at $t_{\mathrm{CL,max}}$. The sample mean of the cloud, indicated by the lines, lies at $-0.22(03) - i\,0.20(04)$ with the standard deviation of the mean obtained through Jackknife resampling and indicated by the shaded areas. The ellipse shows the principal standard deviations of the point cloud, i.e. the square roots of the eigenvalues of the covariance matrix of the cloud, with the semiaxes being the standard deviation values in length oriented along the eigenvectors of the covariance matrix. The simulation was carried out with $\lambda = \beta\mu = 1.0$ with $C_I = 150.0$ resulting in a lattice with $N_\tau = 150$.

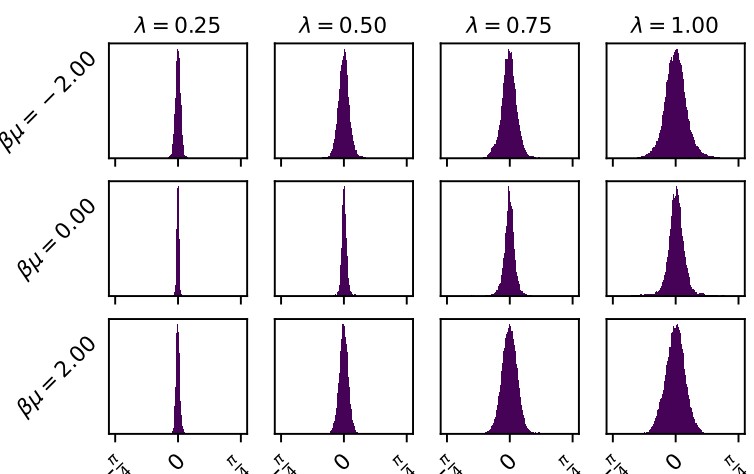

Figure 4: Distribution of the complex argument of the path integral weight $e^{-S_{\mathrm{B}}}$ in the CL sampling process. Samples have been taken with a CL-time spacing of $\delta t_{\mathrm{CL}} = 0.05$, up to a maximum CL time of $15\,000$, where the first $5\,\%$ of the samples have been omitted as "warm up". Here, the size of the lattice is determined by $C_I = 100$, resulting in lattice sizes of up to $N_\tau = 100$ for $\lambda = 1.0$.

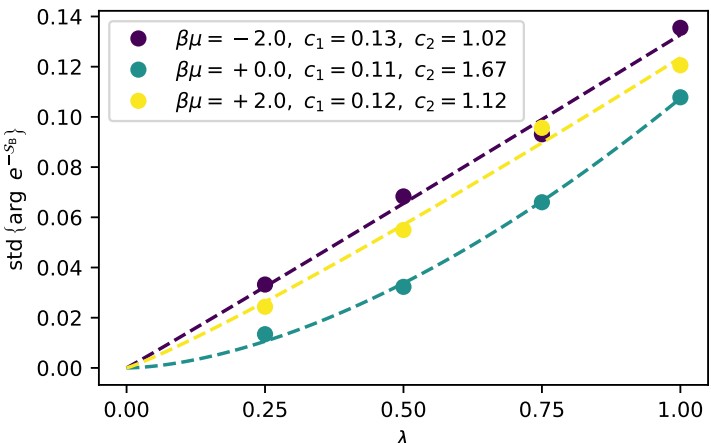

Figure 5: Sample standard deviation of the samples shown in Fig. 4. The dashed lines show fits of the data to the model in Eq. (65) and the fitted parameters are shown in the legend. The width of the argument distribution increases with increasing interaction with an exponent that depends on the chemical potential $\beta\mu$.

with increasing interaction strength, which may not come unexpected. This is illustrated in Fig. 5, where the standard deviations of the phase distributions from Fig. 4 are shown as a function of the interaction parameter $\lambda$ together with a fit of the data to the following model:

$$f(\lambda) = c_1 \cdot \lambda^{c_2} \,. \tag{65}$$

We have chosen this model because the system is known to be free of a sign problem in the non-interacting case associated with $\lambda = 0$ and $f(\lambda) = 0$ coincides with the expected distribution width of zero.

## 5 Conclusions

In this work we successfully developed a lattice formulation of the pairing field formulation of two-component Fermi gases interacting via a two-body interaction. We have shown that our formalism leads to the well-established continuum theory in the limit of large lattices. With our lattice formulation at hand, we employed the CL approach for the computation of densities for a $0 + 1$-dimensional systems. We chose such a low-dimensional model since exact results are available to benchmark our approach. Moreover, studies of low-dimensional models may be viewed to be computationally less intense, at least at first glance. In any case, the use of the CL approach is convenient to deal with the sign problem which is present in our approach even in the absence of, e.g., spin and mass imbalances. In the non-interacting limit, our simulation reproduces the analytic solution exactly. Very importantly, in the phenomenologically relevant interacting case, the results from our stochastic approach are found to be in excellent agreement with the analytic solution, providing a proof of concept for this formalism. Note that we have also analyzed the sign problem in our studies which we found to be mild, at least for all coupling strengths considered in our present work.

For a study of the phase structure and critical behavior of, e.g., ultracold Fermi gases, the computation of correlation functions is indispensable. As we have already demonstrated in terms of the expectation value of the pairing field, there is no conceptional issue within our present approach which would hinder the computation of such functions. Of course, a study of the formation of long-range order or at least quasi long-range order is only meaningful

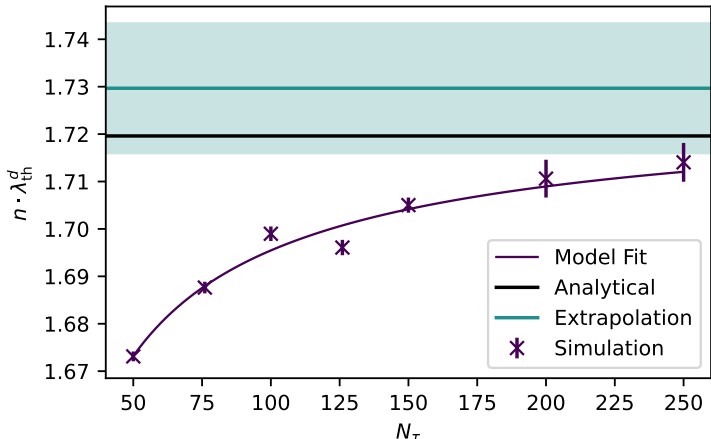

Figure 6: Illustration of the extrapolation of finite-lattice effects in the density observable at the point $\lambda = \beta\mu = 1$. The fit takes the uncertainties of the single simulation runs into account. This results in the shown uncertainty band around the extrapolated value and represents the standard deviation of the estimate.

in higher-dimensional models, $d > 0$. We have already presented the formalism, even for $d > 0$. However, explicit calculations of equations of state and correlation functions for these phenomenologically more models are deferred to future work.

## Acknowledgments

The authors would like to thank the authors of amazing free open-source software that made this work possible. In particular the Julia programming language [95], in which the core of the simulation was written, the Python programming language [96] that was used for data processing and visualization, the matplotlib package [97], the numpy package [98], the scipy [99] package and the sympy package [100] that was used for analytical work in the prototyping stage.

**Funding information** J.B. and F.E. acknowledge support by the DFG under grant BR 4005/5-1. Moreover, J.B. acknowledges support by the DFG grants BR 4005/4-1 and BR 4005/6-1 (Heisenberg program).

## A Lattice artefacts

In our calculations of densities, especially in the regime $\beta\mu \gtrsim -1$, the density exhibits a significant dependence on the size of the lattice that was chosen for the calculations. This dependence is not physical. In order to remove artefacts in the data resulting from the presence of a finite lattice, we calculate observables for multiple lattice sizes and perform an extrapolation to an infinite lattice. For this, we employ a fit to the following empirical model:

$$n_{\text{model}}(N_\tau) = c_1 - c_2 N_\tau^{-c_3} . \tag{A.1}$$

Here, the parameter $c_1$ represents the extrapolated density since

$$\lim_{N_\tau \to \infty} n_{\text{model}}(N_\tau) = c_1 . \tag{A.2}$$

An example for such an extrapolation can be found in Fig. 6.

In the regime $\beta\mu \leq -1$, the densities are largely independent of the lattice size and rather than performing an extrapolation, we calculate the density for a number of reasonably large lattices and use a maximum likelihood estimator to create and estimate for the actual density. Suppose the single lattice calculations result in the densities $\{n_i\}$ for $i \in \{1, ..., N\}$ with standard deviations $\{\sigma_{n_i}\}$, the the maximum likelihood estimator for the density is given by the weighted average

$$\bar{n} = \left(\sum_{i=1}^{N} \frac{1}{\sigma_{n_i}^2}\right)^{-1} \sum_{i=1}^{N} \frac{n_i}{\sigma_{n_i}^2}, \tag{A.3}$$

with the uncertainty

$$\sigma_{\bar{n}} = \left(\sum_{i=1}^{N} \frac{1}{\sigma_{n_i}^2}\right)^{-1/2}. \tag{A.4}$$

This estimator differs from a common average by weighting the densities, such that densities with larger uncertainties contribute less to the estimate.

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
