# Peer review of "A lattice pairing-field approach to ultracold Fermi gases"

_SciPost Physics, doi:SciPost Phys. 16, 091 (2024)_

## Round 2 · Referee Report · Anonymous · 2023-3-2

Strengths
1. Well written
2. Clear
3. Very interesting subject.
Weaknesses
1. Some important points are not addressed.
Report
In this paper the authors formulate a numerical method to study ultra-cold fermionic gases with attractive interactions. To this end, they discretize the Hamiltonian to formulate it on a d-dimensional lattice. They then express the partition function as a path integral over Grassmann variables and a new complex valued auxiliary field which decouples the quartic interaction term into pair creation and destruction operator quadratic terms coupled to this auxiliary field which the authors call a pairing field. The authors then say that the complex valued partition function can be studied numerically using the complex Langevin equation (CLE). To this end, they demonstrate their method by presenting results for the zero-dimensional model (ie one site system).
Since their pairing field is complex, the CLE requires that its real and imaginary parts independently become complex themselves. This requires the stochastic evolution of four fields. Consequently, this method suffers from a sign problem. The authors' main motivation behind this method is that it allows direct access to the pairing order parameter which indicates whether spontaneous symmetry breaking has occurred.
The problem of calculating the BEC order parameter is very interesting and having direct access to it is very helpful. This is an interesting paper, but before I can recommend publication, I would like the authors to address some comments and questions listed here in nor particular order:
1. Since \psi and \psi^\dagger are used in the Hamiltonian to represent the fermionic *operators*, it might be confusing to use the same symbols in Eqs 3, 4, 6 and 7. In these equations, the fermionic operators have been replaced by *Grassmann variables*. In particular, a Grassman variable does not have a Hermitian conjugate, it simply has a conjugate Grassmann variable. The authors use both dagger and conjugate notations in the same equation. This is incorrect.
2. The authors start with the Hamiltonian in the continuum but need to discretize it on a lattice to perform the numerical calculations. This, in fact, leads to a Hubbard model on the lattice. Perhaps the authors did not start directly with the Hubbard model because their main interest is in ultra-cold atoms. However, this method could be of interest in the condensed matter community. In this community the statement, for example, that the particle density (and some other quantities) cannot be calculated directly is rather surprising. Perhaps the authors can add some comments as to how Hubbard model calculations using their method may (or may not) avoid this difficulty.
3. Because of the form of the matrix \Delta (Eq 13), should one understand that the sums over r_i and r_j (eg Eq 12) are effectively over nearest neighbors? It would be good if the authors clarified this.
Now I have several comments that deal with the core of the algorithm presented in the paper.
4. Looking at Eq 7 (or its discretized form), it is clear that the decoupling of the quartic fermionic term is simply a Hubbard-Stratonovitch (HS) transformation where the auxiliary field couples to the fermions off-diagonally (ie to pairing operators) rather than the more often used diagonal coupling (ie to the density operators). The way the authors phrased this gives the impression that this transformation is different. This is a bit misleading.
5. The presentation of this transformation also gives the impression that this is the first time it is being applied. I believe there are several publications dealing with attempts to mitigate the sign problem by using a general HS transformation mixing diagonal and off-diagonal couplings. The authors should make an effort to cite some of them.
6. The earliest off-diagonal HS transformation is in Phys. Rev. B42, 2282 (1990) applied to the Hubbard model with repulsive interactions where it was shown how to decouple the quartic interaction in any spin channel (x,y, or z). Interestingly, in that publication the auxiliary field is *real* not complex, and in fact it is also shown that the same off-diagonal HS transformations can be implemented with a *discrete* auxiliary field, the same as is used in determinant QMC. DQMC simulations were presented in this publication. It appears the authors are not aware of this paper. Looking at Eq 7 of the current paper, I do it is not clear why a complex field is required. It is true that in general the BEC order parameter is complex, but that corresponds to a global U(1) gauge invariance in the model under discussion. Fixing the global gauge by using real auxiliary fields does not cost us any information and would make the CLE much simpler as it now has only two fields instead of four. This could make it more efficient.
7. Concerning the sign problem faced by the authors: They mention that it gets worse as the coupling increases. But how does it behave with decreasing temperature? And do they have any indication how severe it becomes for a multi-site system?
8. Still with the sign problem: I think (I have not checked this, but the authors should) that by using the real auxiliary field as shown in the paper cited in point 6, and applying it to the *attractive* case which they consider here, there will *not* be a sign problem. Then one can use DQMC or a real Langevin equation to study the behavior of the pairing more efficiently than with complex fields. Have the authors tried and discarded this approach?
Requested changes
See full report
Author: Florian Ehmann on 2023-04-18 [id 3595]
(in reply to Report 1 on 2023-03-02)We would like to thank the referee for their careful reading of the manuscript. In particular, we are grateful for their positive report and for stating that our manuscript is "interesting" and "well written" and that it addresses a "very interesting subject".
We have improved and extended our manuscript, following the report of the referee. Below, we comment in detail on the changes made to the manuscript. In particular, we reply to the referee's comments as well as suggestions following the order in which they appear in the report. We hope that the referee deems the present version of our manuscript suitable for publication in SciPost Physics.
1 . The referee is absolutely right. These are unfortunate typos. We corrected the corresponding equations in the revised version, see Eqs. (4) and (7).
2 . Our focus is basically on the description of ultracold atomic Fermi gases in d+1 dimensions and possibly also on extensions of our formalism for a description of nuclear matter. However, we absolutely agree with the referee that the discretized version of our model can be directly related to the Hubbard model.
We have added two paragraphs on this aspect to point the reader to this connection which can be found on page 1 (left column, starting in line 3 from below, "From a phenomenological standpoint, we have ... can be directly related to condensed-matter systems, most prominently to the Hubbard model, see, e.g., Refs. [37, 38].") and on page 3 (right column, line 12-16, "At this point, ... of the present work."). Note that we have also included two new references in our manuscript, see Refs. [37, 38].
3 . The sums in Eq. (12) run over all spatial sites of the lattice. With our concrete choice for the discretization of the spatial derivative, they effectively reduce to an on-site contribution and contributions of nearest neighbors. However, other choices for the discretization of the spatial derivative are possible, e.g., with contributions from neighbors farther away and no contributions from nearest neighbors at all. Therefore, we opt to keep the sums general.
We clarified this aspect below Eq. (13) in the revised version of the manuscript, see page 13 (right column, from line 2-11, "With our concrete choice ... their general form.").
4 . We do not intend to mislead the reader here. Of course, we simply use a specific type of Hubbard-Stratonovich transformation after all. We only intend to emphasize that the Hubbard-Stratonovich transformation employed in our manuscript couples to a bosonic field associated with a pair of one spin-up and one spin-down fermion, which is phenomenologically distinct from the density field. Moreover, as we mentioned in the discussion of the Hubbard-Stratonovich transformation underlying our present work, this transformation is basically the standard Hubbard-Stratonovich transformation employed in continuum studies of ultracold gases.
We extended our discussion at two points to address this issue: On page 2 (left column, line 3-4 of Sect. II, "It is based on ... in the literature."), we have added a first brief comment. On page 5/6 (starting on page 5, right column, line 10 from below to line 12 of the left column of page 6), we extended our discussion to put our present work in context with other works where Hubbard-Stratonovich transformations are used. Here, we als included two new references, see Refs. [76, 77] of the revised version of our manuscript. Note that this change of the comment is also partially related to the comments 5 and 6 of the referee, see below.
5 . and 6 . Since we believe that comments 5 and 6 of the referee are related, we would like to reply to them simultaneously:
a) First of all, we would like to thank the referee for pointing us to Ref. [Phys. Rev. B 42, 2282 (1990)] which we included in the revised version, see Ref. [76] of the revised version. We explicitly mention this article in the discussion of the Hubbard-Stratonovich transformation (HST) underlying our present work, see page 5/6 (starting on page 5, right column, line 10 from below to line 12 of the left column of page 6).
b) The introduction of the complex (pairing) field associated with our Hubbard-Stratonovich transformation generates a sign problem. To deal with this problem, we utilize the CL approach, which requires to complexify the real and imaginary parts of the pairing field.
c) The authors of the aforementioned work (see Ref. [Phys. Rev. B 42, 2282 (1990)]) use a purely spatial representation of the problem whereas we are interested in developing an approach which is built on a spacetime representation of the auxiliary field. We also used the purely spatial representation in the past where we employed an auxiliary field which, loosely speaking, carries the quantum numbers of the fermion density, see Refs. [76] of the revised version of our manuscript. The spacetime representation is more familiar in, e.g., high-energy physics because space and time have to be treated on an equal footing. In practice, our auxiliary field is related to the anomalous density which may therefore also be viewed as a field associated with a pair of one spin-up and one spin-down fermion.
In any case, the motivation for our choice of the Hubbard-Stratonovich transformation is twofold: First, we are interested in the development of a formalism which "naturally" allows for a computation of, e.g., spacetime-dependent spectral correlation functions of the pairing field. Second, we would like to establish a formalism which is as close as possible to the established existing continuum approaches to ultracold Fermi gases which may help to facilitate comparisons of results from lattice and continuum approaches in the future.
d) Formally, we employ a complex spacetime dependent Hubbard-Stratonovich field because, looking at, e.g., Eq. (6), there is a priori no reason to assume that the full dynamics can be captured by only using the real part of the Hubbard-Stratonovich field. Also, in our CL equations, we presently do not see indications that the dynamics in the CL runs is only driven by either the real or the imaginary part of this field. We would like to add that it may be an interesting option in the future to parametrize our complex-valued spacetime dependent Hubbard-Stratonovich field by its spacetime dependent modulus, which is real-valued by construction, and a spacetime dependent phase factor. Loosely speaking, this phase factor describes the dynamics of the Goldstone mode in cases where the ground state is governed by the formation of a superfluid.
7 .
a) Effectively, decreasing the temperature has the same effect as increasing the coupling in our model, see our definition of the dimensionless coupling $\lambda$ between Eqs. (16) and (17).
b) Up to now, we have only run first tests for the 1+1d case but not yet detailed calculations of observables.
8 . As shown in Ref. [Phys. Rev. B 42, 2282 (1990)], for certain combinations of operators, we agree that it is possible (as shown by that work) to represent the interaction using a purely real field. However, since it produces a sign problem either way (also as shown in the aforementioned work), it is required to use an approach which can handle a sign problem (such as CL) anyway. In any case, we agree that it may be interesting in the future to employ the Hubbard-Stratonovich transformation discussed in Ref. [Phys. Rev. B 42, 2282 (1990)] for numerical studies and compare the resulting data with our present results. However, this is beyond the scope of this work.
Additional changes:
a) We corrected a typo in Eq. (28).
b) We corrected a typo in the definition of the thermal wavelength, below Eq. (61).
c) We corrected a typo in Eq. (64).
d) We corrected a typo in Eq. (A3) [and correspondingly in Eq. (A4)].
e) We corrected a typo in the caption of Fig. 3.

---

## Round 2 · Referee Report · Anonymous · 2023-5-22

Report
The authors propose a formalism for computing specific observables in Quantum Monte Carlo simulations of strongly correlated fermionic theories. In this formalism, a specifically tuned Hubbard-Stratonovich transformation is employed to couple the auxiliary field with the fermionic bilinear terms relevant for the calculation of observables.
Within this framework, the average value of the fermionic bilinear corresponds to the average of the auxiliary field. Consequently, the computation of averages for products of fermionic propagators can be replaced by evaluating the correlations of the auxiliary fields.
In principle, this is a known feature of the Hubbard-Stratonovich (HS) transformations, see e.g. Eq. 13 in arXiv:2012.11914.
However, this exact version of the HS transformation for Hubbard-type interactions was rarely used in actual simulations (though it was actually discussed e.g. in arXiv:1403.5680), mainly due to the sign problem hindering our ability to simulate large systems.
And here, in my opinion, lies the most intriguing aspect of the paper. It is a successful utilization of the Complex Langevin (CL) algorithm to overcome the sign problem. Currently, the method has been tested only on a (0+1)D system, where analytical results are available for comparison. This comparison has validated the accuracy of the CL results. However, the application to (d+1)D systems is left for future papers.
Despite the confirmation of CL computations through exact analytical results, I would still like to request the authors' comments on a well-known issue associated with CL methods. Namely, when the trajectory encounters a singularity in the drift term, often caused by the vanishing determinant in fermionic theories, systematic errors can arise that are challenging to correct. Papers such as arXiv:1508.05252 and arXiv:1701.00986 discuss this issue in detail, and it is shown that the success in (0+1) dimensions does not guarantee the effectiveness of CL algorithm in higher dimensions. Therefore, it is necessary to provide some insights into the validity of the CL method in higher dimensions for this particular model. Additionally, it would be interesting to understand why CL performed so well in the (0+1)D case. Could it be attributed to the absence of singular points in the drift term?
As a side note, there exists a general framework that categorizes various fermionic theories prone to the sign problem under a unified classification scheme. The authors may find it beneficial to look at the following papers: arXiv:1408.2269 and particularly arXiv:1601.05780. It would be interesting to determine the specific symmetry class to which their model belongs taking into the specific channel of HS transformation used by authors.
In principle, it is possible that the sign problem could vanish in the case of different pairing, rendering the advantage of direct access to fermionic observables through the bosonic pairing field obsolete.
This concern is particularly valid because the advantage of computing fermionic observables through bosonic correlators may be not that significant. In fact, in the case of Determinantal Quantum Monte Carlo (DQMC), direct access to the fermionic propagator is available during the auxiliary field update process. Even if the fermionic determinant is computed using stochastic estimators, like in conventional Hybrid Monte Carlo, the cost of calculating observables is still lower compared to updating the field configurations. It would be interesting to read the authors' comments on this potential possibility.
In general, I find the paper to be an interesting test of the Complex Langevin algorithm applied to a non-trivial, strongly correlated fermionic model. Given this, I believe that the paper should be published once the authors address the aforementioned points.

---

## Round 4 · List of Changes

We have included remarks and references in the manuscript to reflect these comments:

1.- We added a reference to arXiv:1403.5680 on page 2 (left column, line 16-20, "We add that our ... see also Ref. [52] for a more general discussion.").

2.- In the right column of page 5, we have slightly adapted the sentence right below Eq. (36) and supplemented it with a footnote (see new footnote #3).

3.- We have adapted our general comments on the sign problem on page 6 (left column, line 4-14, "In this context, it is also ... with, e.g., Ref. [52]"), where also included five new references, see Refs. [52, 78-81].

4.- We have extended our discussion at the end of Subsec. IV.D., where we now comment on potential problems associated with singularities in the drift (see page 10, from line 9 in the left column to line 5 in the right column, "Beyond the expected ... treatment with the complex Langevin method."). Here, we also included six new references, see Refs. [89-94].

Additional change:

We have rephrased the sentence right above Eq. (32) to amend the column breaks on page 5.

---

## Editorial Decision

published